# Mechanical and Microstructural Properties of Ordinary Concrete with High Additions of Crushed Glass

**DOI:** 10.3390/ma14081872

**Published:** 2021-04-09

**Authors:** Cherif Belebchouche, Karim Moussaceb, Salah-Eddine Bensebti, Abdelkarim Aït-Mokhtar, Abdelkader Hammoudi, Slawomir Czarnecki

**Affiliations:** 1Départment de Génie Civil, Faculté des Sciences de la Technologie, Université Frères Mentouri Constantine 1, Constantine 25000, Algeria; s_bensebti@yahoo.fr; 2Département de Technologie, Faculté de Technologie, Université Abderrahemane Mira de Bejaia, Bejaia 06000, Algeria; articlesmoussaceb@yahoo.fr (K.M.); abdelkader.hammoudi@univ-bejaia.dz (A.H.); 3Laboratoire des Sciences de l’Ingénieur pour l’Environnement UMR CNRS 7356, Université de La Rochelle, 17042 La Rochelle, France; karim.ait-mokhtar@univ-lr.fr; 4Faculty of Civil Engineering, Wroclaw University of Science and Techology, Wybrzeze Wyspianskiego 27, 50-370 Wroclaw, Poland

**Keywords:** ordinary concrete, crushed glass waste, mechanical strength, hydration degree, porosity, sustainable materials

## Abstract

This study investigates the use of crushed glass waste as partial cement replacement in ordinary concretes. Six concrete mixes were designed and prepared: a reference without substitution and five substitution percentages of crushed glass waste ranging from 5% to 25%. The made concrete mix design underwent different tests, namely: slump test, mechanical strength, thermogravimetric analysis (TGA), X-ray diffraction (XRD), Brunauer–Emmett–Teller (BET) determination and finally, water porosimetry, in order to evaluate the influence of the use of crushed glass waste on the properties of fresh and hardened concrete. Mechanical strengths results show that the use of 15% of the crushed glass waste improves the mechanical strength. TGA analysis confirms this result by highlighting a higher hydration degree. The latter contributes to the reduction of the porosity and, consequently, the mechanical strength increases. Also, it can be caused by the increasing amount of chromium which, if added a little, accelerates the hydration of C3S and leads to an increase of the mechanical strength. The BET technique and porosimetry tests showed that the use of crushed glass waste reduces the global porosity of concrete. This is due to the filling effect of the glass powder.

## 1. Introduction

The cement industry is one of the most polluting sectors with a contribution of nearly 7% of global CO_2_ emissions [1,2]. This poses a threat to human health and the environment. The CO_2_ emission is mainly due to the decomposition of limestone [3,4]. The use of additions as a partial replacement for cement seems to be an effective technique to reduce its environmental impact [5,6,7,8,9]. Also it reduces costs, preserves natural resources, saves energy and reduces the volume of waste [10,11,12].

The use of wastes for partial replacement of cement, such as ground granulated blast furnace slag, fly ash, marble powder and glass powder have been studied by many researchers [13,14,15]. However, glass can be considered as the most appropriate as cement substation, due to its chemical composition and physical properties [16,17,18,19,20] as well as its abundance in the landfills in large quantities. In Algeria, almost 170,000 tons/year of glass are thrown in the environment, knowing that a glass bottle takes at least 4000 years to be degraded [21,22]. 

Several scientists investigated the use of glass powder in some cementitious materials. It has been noticed that the incorporation of glass powder in self-compacting concrete considerably improves their performances with respect to the aggressive media due to its pozzolanic potential [6]. It was also reported that the use of 20% of glass powder increase the compressive strength of concrete [7]. Moreover, using 10% of glass powder as cement replacement improves concrete compressive strength, tensile strength, absorption, voids ratio and density; however, more than 15.0% as cement replacement decreases the 28-day concrete compressive strength [23]. Also usage of glass powder in concrete improves the compressive strength and decreases the porosity at 28 days [24]. In Table 1, there are exemplary studies presented, in which waste glass powder was used as an eco-friendly concrete admixture.

According to the literature survey, there are various ideas of using waste glass as a by-product in cementitious composites. However the knowledge of the behavior of hardened concrete, with glass powder used as a cement substitute is still full of the research gaps. Even if the glass powder is replacing cement, very often there is only one dosage of waste material tested [19] or the differences in mixtures of cementitious composites by using too many different admixtures are significant so that the most effective mixture in terms of dosage of waste glass powder might be omitted [18]. There is also a lack of deeper understanding in most of aforementioned works by creating models of correlating the compressive strength with other investigated mechanical-chemical properties. Thus, the main research goals presented in this work are to design a sustainable ordinary concrete containing a waste glass powder (GP) as partial cement replacement, differing the dosage of GP from 5% to 25%. Based on the performed laboratory tests of compression strength the most efficient concrete mixture was selected. The application goal is to design a product made of cement and waste glass powder which is attractive from economical and sustainable points of view, and can be used in cementitious composites. Thermogravimetric analyses, X-ray diffraction analyses and water porosimetry tests were performed. After the tests, the properties evaluated using these methods were correlated with the compressive strength. That allows a deeper understanding of the ongoing processes during concrete hardening. Also, in order to meet the application goal the calculation of the price reduction using waste glass powder as a cement substitute was performed. Based on the ranking method the most efficient mixture of cementitious composite with waste glass powder was chosen. These will fill the research gap and may convince other researchers to reduce the amount of cement substituting it by waste glass powder more often. 

## 2. Materials and Methods

### 2.1. Materials and Mix Design

A Portland cement of type of CEM II 42.5 from GICA factory, according to European standard NF-EN 197-1, was used. This cement comes from the El-Hamma region of Constantine, Algeria, and its grain-size distribution and physico-chemical characteristics are illustrated in Figure 1 and presented in Table 2, respectively. Calcareous crushed aggregates from the National Company of Aggregates (NCA) of El-Khroub region of Constantine, Algeria, were used. They are of three granular classes: sand 0/3 mm, gravel 3/8 mm and gravel 8/16 mm. The glass powder used was obtained by crushing glass bottles recovered from public landfills; its grain-size distribution is illustrated in Figure 1a. The bottles used have the same nature and green color. From the results of grain-size distribution, it can be observed that the glass powder is finer than the cement. In the Figure 1b,c the particle shape and the major components of glass powder used are presented, obtained by the scanning electron microscope “SEM” coupled with energy-dispersive spectroscopy “EDS” Philips/FEI XL 30S FEG Chatsworth, CA, United States. According to this figure, the glass bottles, after crushing, show angular shapes and it is possible to observe an almost homogeneous size distribution for a range of sizes from 0–50 μm. Also, we note the presence of silicon, oxygen, sodium, calcium, aluminium, magnesium, potassium and iron. The elementary chemical composition of the glass powder, obtained by X-ray fluorescence and energy-dispersive X-ray spectroscopy (EDS), is reported in Table 2 and Table 3. Cement and glass powder samples were analysed in triplicate. Results of Table 2 and Table 3 show the presence of silica in a large proportion, which may give rise to a possible pozzolanic effect which occurs during cement hydration [26].

The Portland cement-based concrete had an expected compressive strength of 30 MPa and a slump of 100 mm. The glass powder was added as a replacement of a part of the cement from 5% to 25% and a reference without glass powder (see Table 4). The concrete without glass powder serves as a reference for comparisons. 

### 2.2. Experimental Methods

#### 2.2.1. Slump Test

The fresh concrete workability, measured by its slump, was monitored using Abrams cone according to EN 12350-2 standard [27]. The steel slump cone is placed on a solid, impermeable, level base and filled with the fresh concrete in three equal layers. Each layer is tamped 25 times to ensure compaction. The third layer is finished off level with the top of the cone. The cone is carefully lifted up, leaving a heap of concrete that slumps slightly. The upturned slump cone is placed on the base to act as a reference, and the difference in level between its top and the top of the concrete is measured and recorded to the nearest 10 mm to give the slump of the concrete.

#### 2.2.2. Mechanical Strength

Compressive strengths were measured using a hydraulic press of type 65-L11M2 according to the NF EN 206 standard [28]. The compressive strength was calculated in accordance to the Equation (1):(1)$$f_{c}=\frac{F}{A}$$
where: *F*: destructive force measured [kN];*A*: area of the sample [mm^2^].

#### 2.2.3. Thermogravimetric Analysis (TGA)

The TGA is used to study the influence of the addition of glass powder on the hydration degree of the studied materials. The device used in our study is the SETARAM type TG-DTA 92. The hydration degree α (t) of cement is then calculated from Equation (2) [29,30]:(2)$$\alpha \;\left(t\right)=\frac{m_{el}\left(t\right)}{wel\left(\infty \right)\times m_{c}}\times 100$$
(3)$$m_{el}(t)=\left|\Delta m_{145{}^{\circ}C\to 1100{}^{\circ}C}(t)\right|-\left|\Delta m_{600{}^{\circ}C\to 800{}^{\circ}C}(t)\right|+m_{d,145{}^{\circ}C\to 1100{}^{\circ}C}(t)-m_{c}\times LOI$$
(4)$$m_{c}=\frac{m_{sample}}{\left(1+\frac{S}{C}+\frac{A}{C}+\frac{W}{C})\times (1+LOI\right)}$$
(5)$$wel(\infty )=0.24C_{3}S(\%)+0.21C_{2}S(\%)+0.4C_{3}A(\%)+0.37C_{4}AF(\%)$$
where: *m_el_* (*t*): the water mass related to the cement at the instant « t » [kg];*mc*: the anhydrous cement mass added to the sample [kg];Δ*m*_145°C__→1100°C_(*t*): sample mass loss between 145 °C and 1100 °C in [kg];Δ*m*_600°C__→800°C_(*t*): sample mass loss between 600 °C and 800 °C in [kg];*m_d,_*_145°C__→1100°C_(*t*): instrument drift between 145 °C and 1100 °C in [kg];*m_c_*: the anhydrous mass of cement introduced into the sample in [kg];*LOI*: ignition loss of the anhydrous cement in [%];*m_sample_*: Initial mass of the sample in [kg];*W/C*: water/cement ratio [%];*S/C*: sand/cement ratio [%];*A/C*: mineral addition/cement ratio [%];*Wel*(∞): amount of water required for complete hydration of the cement [%]. This amount of water is estimated according to the composition of the cement by Bogue equations.

The mass content of Portlandite Ca(OH)_2_ (g/g of cement) present in the cement paste is calculated from the mathematical expression (6) [26]:(6)$$m_{\mathrm{Ca}{\left(\mathrm{OH}\right)}_{2}}\left(t\right)=\frac{\left|\Delta m_{400{}^{\circ}C\to 500{}^{\circ}C}\left(t\right)\right|+m_{d,400{}^{\circ}C\to 500{}^{\circ}C}\left(t\right)}{m_{c}}\times \frac{M_{\mathrm{Ca}{\left(\mathrm{OH}\right)}_{2}}}{m_{H_{2}O}}$$
where:Δ*m*_400°C__→500°C_(*t*): sample mass loss between 400 °C and 500 °C in [kg];*m_d,_*_400°C__→500°C_(*t*): instrument drift between 400 °C and 500 °C in [kg];$M_{\mathrm{Ca}{\left(\mathrm{OH}\right)}_{2}}$: molar mass of Portlandite in [kg/mol];$m_{H_{2}O}$: molar mass of water in [kg/mol]. 

#### 2.2.4. X-ray Diffraction (XRD) Analysis

A diffractometer type X’PERT PRO PANalytical was used to analyze the mixes studied in the form of fine powders < 100 µm. The measurement conditions are as follows: 10° < 2*θ* < 80° and a step of 0.017°. Then, the diffractograms obtained were treated using the software X’Pert High Score to determine the crystalline phases of the sample analyzed, especially Portlandite Ca(OH)_2_.

#### 2.2.5. Water Porosimetry

Open porosity is considered as an essential parameter with respect to the concrete’s durability. Water porosimetry was carried out according to the AFPC-AFREM procedure [31]. To do this, three samples were tested for each mixture at 28 days of cure. This method consists of weighing the samples in different states. For this, the sample is placed in a vacuum desiccator for 4 h, then the samples are immersed in water, still under vacuum. After 24 h at atmospheric pressure, the sample is weighed for the first time in water using a hydrostatic scales, the mass of the submerged sample is then obtained “*M_sub_*”. Always saturated, the samples are then weighed in air and thus the mass of the saturated samples obtained “*M_air_*”. At this stage, the sample is placed in an oven at a temperature of 105 °C, until the mass of the sample stabilizes, which will then be assumed to be dry. If this mass stabilization is achieved, then the mass of the dry sample is obtained “*M_dry_*”. The porosity “ε” is then calculated as follows:(7)$$\varepsilon =\frac{M_{air}-M_{dry}}{M_{air}-M_{sub}}\times 100$$

#### 2.2.6. Porosimetry by Desorption Isotherms (Brunauer–Emmett–Teller (BET))

Sorption measurements combined with Brunauer–Emmett–Teller (BET) theory allow the determination of the specific surface area and pore size distribution from desorption isotherms. The test involves placing a sample in a closed volume, with a certain gas pressure, where an increase in the mass of the solid is observed accompanied by a decrease in the gas pressure. This increase in mass, which is proportional to the quantity of gas adsorbed, depends on the temperature T, on the pressure of the gas P and on the chemical nature of the solid/gas pair. During the test, the amount of adsorbate retained on the surface of the solid equilibrates as a function of the pressure of the gas (Activity in the gas phase). The representation of the set of equilibrium states corresponding to pressures between 0 and the saturated vapor pressure (P0) is called the sorption isotherm. This curve is characteristic of the adsorbent/adsorbable couple studied. Before starting the analysis, the samples are degassed at 200 °C for 1 h and then weighed. For all samples, the measurements were carried out with nitrogen, the latter acts as the adsorbable at its normal liquefaction temperature. The activity in the gas phase expressed in grams or mol of adsorbent per unit mass of solid adsorbent is represented as a function of the relative pressure P/P0. The interpretation of adsorption-desorption isotherms by the BET method makes it possible to determine the average pore size distribution and the specific surface area of the sample.

An estimation of pore size distribution can be obtained from desorption isotherms by using Kelvin formula and Laplace equation, which aim at determining the average pore diameter [32,33].
(8)$$d=-\frac{2\gamma Vcos\theta }{RTln(P/P_{0})}$$
where: d: the average pore diameter;*R*: the gas constant;*V*: nitrogen molar volume;*T*: nitrogen temperature;*γ*: the surface tension of nitrogen;*θ*: the contact angle.

Then, the specific surface area is obtained according to the following formula (9):(9)$$\mathrm{As}=-\frac{n_{m}a}{m}\times N_{A}$$
where: As: the specific surface area;*n_m_*: the number of moles of adsorbate in a monolayer;*m*: the sample mass in grams;*a*: the cross-sectional area of the adsorbate molecule of nitrogen;N_A_: Avogadro’s number.

## 3. Results and Discussion

### 3.1. Slump Tests of Fresh Concrete

Slumps of fresh concrete with Abrams cone are presented in Table 5.

According to the results shown in Table 5, we can note that the reference concrete presents an important slump than those containing glass powder. The classes of the slump obtained are: S3 “very plastic concrete” for reference concrete BT and B5%, S2 “plastic concrete” for the concretes B10%, B15%, B20% and B25%. The evolution of the measured slumps is due to the increase in amount of glass in the mixture consumes a part of water due to its high specific surface determined by BET theory, which is in the order of 4530 cm^2^/g, compared to that of cement (3310 cm^2^/g) and consequently leads to low slumps. Also with the increase in the glass volume in the mixture results in an important frictions between solid grains due to the angular shape (see Figure 1b) of the glass powder particles, thereby reducing the concrete slump.

### 3.2. Compressive Strength of Hardened Concrete

Compressive strength test was performed after 7 and 28 days in order to compare the conformity of concretes studied with the requirements and construction standards. The failure models and the test stand are presented in Figure 2. In Figure 3 the results of the test as a compressive strength are shown.

In general, the failure models are similar for all of the samples and are in accordance with the literature. It appears that the strength of the studied concretes increases with the increase in the amount of the glass powder to a percentage of 15%. 

The latter is more dense [34] and contributes to the reduction of the porosity and, consequently, the mechanical strength increases. Moreover the increase of the amount of glass powder in the mixtures increases the amount of chromium. The latter accelerates the hydration of C3S and leads to the formation of hydrates which give the concretes better mechanical strength [35,36].

The high specific surface of glass powder contributes to the filling effect (see Figure 1). The latter reduces significantly the concrete porosity and leads to an increase of its strength [35].

An effect of addition more than 15% of glass powder is the lowering compressive strength below the value obtained for BT sample. This can be explained by the reduction of cement amount in the material, decreasing the rate of hydrates (C-S-H and portlandite) that provide the concretes strength. Also, it can be caused by the excess of the amount of chromium supplied by the glass powder. The addition of chromium to cementitious materials with moderate quantities causes an increase in the mechanical strength. When the amount of chromium exceeds a certain percentage, it causes an opposite effect and consequently slows down the hydration of the cement grains [35,36].

### 3.3. Hydration Degree

To better illustrate the improvement of strength of mixed concrete B5%; B10% and B15% comparing to BT; B20% and B25%, TGA analyses were performed. Hydration degrees and amount of portlandite obtained from thermogravimetric analysis at 28 days of cure are given in Figure 4 as a function of the amount of glass powder.

The histogram of hydration degrees given in Figure 4a coincides with that of mechanical strengths (see Figure 3). It can be seen that the hydration degree of materials containing percentage less than 15% of glass powder (i.e., 5%, 10% and 15%) is higher than that of reference concrete. However a slight increase in the hydration degree was observed in B5%, B10% and B15% despite the decrease of the amount of cement in the mix design. From TG curves (Figure 4b) and the histogram of the amount of portlandite (Figure 4c) estimated at 28 days of cure, despite the decrease in the amount of cement in B20% and B25%, it was observed that the mass loss of portlandite between 400 and 500 °C for B15% is almost of the same order as those for B20% and B25%. This confirms that silica contained in the amorphous glass reacts with portlandite formed during the hydration of cement phases (C3S and C2S) and forms C-S-H [17,34]. This reaction reduces the porosity and increases the concrete strength. Also, it can be explained by the increase of chromium amount (see Table 2) in the mixtures because the latter accelerates the hydration of cement particles by the formation of hydrates which give the concretes better mechanical strengths [35,36].

From a percentage of 15%, it has been observed that the hydration degree and compressive strength decreases in spite of the increase in the glass powder amount. It can be seen in Figure 5 that the compressive strength of concrete is also affected by the hydration degree. It has been proved by the relatively high value of linear coefficient of correlation equal to 0.8959. Such behavior of hardened concrete is essentially due to the decrease in the cement amount in the mixture which becomes unsatisfactory to ensure a large quantity of hydrates and as a result, the porosity of the concrete increases and the mechanical strength decreases [4]. 

Also, it can be caused by the excess of the amount of chromium which slows down the hydration of the cement grains when its amount exceeds a certain percentage [35,36]. To better illustrate what happened in the structure of the formulations studied, XRD analyses (see Figure 6) were carried out on the reference concrete BT, the B15% which exhibited better compressive strength and the B25% with lowest compressive strength compared to the formulations studied.

The spectrograms obtained were treated using X’Pert High Score software to determine crystalline phases using ICDD “International Centre for Diffraction Data” cards (See Table 6).

The spectrograms obtained from XRD analysis show the presence of portlandite (Ca(OH)_2_) and ettringite that are the result of hydration reactions. Also the presence of anhydrous cement grains in the mixtures. The results of XRD measurements were analysed using the Rietveld method [37]. It was noticed, that when amount of glass powder increases, amount of larnite (C2S) and calcium silicate (C3S) tends to decrease in B15% compared to that of BT. C2S + C3S decreased about for 17.9% from 42.6% in BT to 24.7% in B15%. Decreased amount of C2S and C3S is probably related with a better solubility of the cement phase. Pozzolanic reaction by portlandite with silica of glass powder, probably, had the biggest influence on decreased intensities of portlandite peaks [38]. The amount of portlandite decreased about for 5.7% from 9.8% in BT to 4.1% in B15%. This confirms the results of TGA analysis and justifies the pozzolanic potential of glass powder [17,34]. The uncrystallized zones which are observed between peaks in the mixes BT and B15% may be attributed to the amorphous gel of C-S-H [39,40] and/or to the amorphous silica of glass powder [12] which are not detectable by XRD.

According to XRD results, we can conclude that, the addition of 15% of glass powder in concrete decreases considerably the amount of portlandite due to pozzolanic reactions of silica and portlandite. This phenomenon is considered as an advantageous gain limiting the concrete degradation after its curing toward sulphate attacks and chloride ions [41].

### 3.4. Evaluation of Porosity Accessible to Water and Average Pore Size

The specifications for the formulation of concretes take into consideration the mechanical properties as well as the fresh properties of these materials. The material’s porosity is the first indicator of concrete durability towards external aggressions. Figure 7 shows the evolution of porosity and average pore size of the different formulations studied, at about 6 months of age, according to their formulations.

From the results illustrated in Figure 7, it is noticed that the porosity varies in the same way as the average pore diameter. This result coincides well with those of mechanical strength, higher is the water porosity and weaker is the mechanical strength. These results confirm also the optimal of 15% glass powder substitution, which is also presented in Figure 8. According to this figure, the values of the compressive strength strongly correlate with the water porosity percentage. It is proved by the very high value of the linear coefficient of correlation r^2^ equal to 0.9835.

The average pore diameter of the reference concrete BT is almost 162 nm, whereas that of concrete containing 15% of glass powder is 95.3 nm. This is due to (i) the pozzolanic potential of glass powder to produce more C-S-H which densifies the structure of the concrete and improves its connectivity, and (ii) to the filling effect of the glass powder particles due to its high fineness compared to that of cement, thereby reducing capillary pore size and concrete porosity. Beyond a substitution percentage of 15%, a significant increase in the average pore diameter accompanied by an increase in porosity is observed. This is mainly due to an insufficient amount of cement which serves to maintain the compactness of the mix from the initiation of hydration process [42].

## 4. Ranking Method for Choosing the Most Effective Cementitious Composite Mixture

In order to choose the most effective cementitious composite mixture the ranking method has been selected. The two analysed properties were cost of the material and compressive strength of concrete for all of the investigated mixtures including the reference sample. The assumption of the price of concrete was made based on average market values of concrete in European Union which is about 100 Euro/m^3^. Most of this price is the cost of the cement which is about from 30% to 35% of this price (c.a. 30 Euro/m^3^). The cost of the waste glass powder has been assumed as 10% of replaced cement price which is about 3 Euro/m^3^. The calculation of price of prepared concrete mixtures with waste glass powder and the decision ranks of the compressive strength and the cost of the cementitious composite mixtures are presented in Table 7.

Analysing the results presented in Table 7. It can be seen that the most effective in terms of quality and price, according to the ranking method is the mixture BT15% containing 297.5 kg of cement, 52.5 kg of waste glass powder, 175 kg of water and 1661 kg of aggregate per 1 m^3^.

## 5. Conclusions

Presented in this article the laboratory tests proved that there is possible to obtain a mixture by incorporating the waste glass powder into the concrete, which has positive effect in terms of physico-mechanical properties. It has been proved by performing the analyses of the compressive strength, hydration degree and porosity. It was observed that both properties (hydration degree and porosity) strongly correlate with the compressive strength of cementitious composite with waste glass powder. Also the second goal was achieved, because using for this purpose waste glass powder obtained in the process of utilization of the glass bottles decreases the price of the concrete and it is environmentally friendly. It has been evaluated using the ranking method and proved the positive effect of the economical aspect, in terms of quality/price, of using waste glass powder as an admixture.

Results obtained through this study, highlight the following main points:The incorporation of glass powder directly influences the mechanical strength at 7 and 28 days of age. Replacing cement with dosage of 5%, 10% and 15% of cement by waste glass powder increases the value of compressive strength obtained at 7 and 28 days of age. Replacement of 15% cement by glass powder shows the highest compressive strength in comparison to the other substitution percentages. This is may have considered as the optimal dosage;TGA analysis shows that cement replacement by glass powder with dosage 15% and less increases the hydration degree of concrete. This is due to a small amount of chromium oxide contained in glass powder. These compounds accelerate the hydration reactions of cement particles. However, increasing the dosage of waste glass powder more than 15% has an opposite effect and the rapid decrease of the hydration degree was observed;XRD analysis of concrete mix design B15% highlighted the decrease of the intensities of portlandite peaks and confirmed the pozzolanic potential of glass powder that is considered as an advantage to limit the concrete degradation after its curing toward sulphate attacks and chloride;The results of BET analysis have shown that the cement replacement by glass powder up to 15% has reduced considerably the average pore diameter and therefore the concrete porosity.

It would be desirable to complete this study by further investigations on the microstructure morphology of the obtained concrete and its relationship with permeability, which governs aggressive agent penetration within the concrete [43]. This would allow us to highlight in more depth the effect of the incorporation of crushed glass in cement-based materials. In addition, studying the properties, variability of such materials [18] will complete this first approach.

## Figures and Tables

**Figure 1 materials-14-01872-f001:**
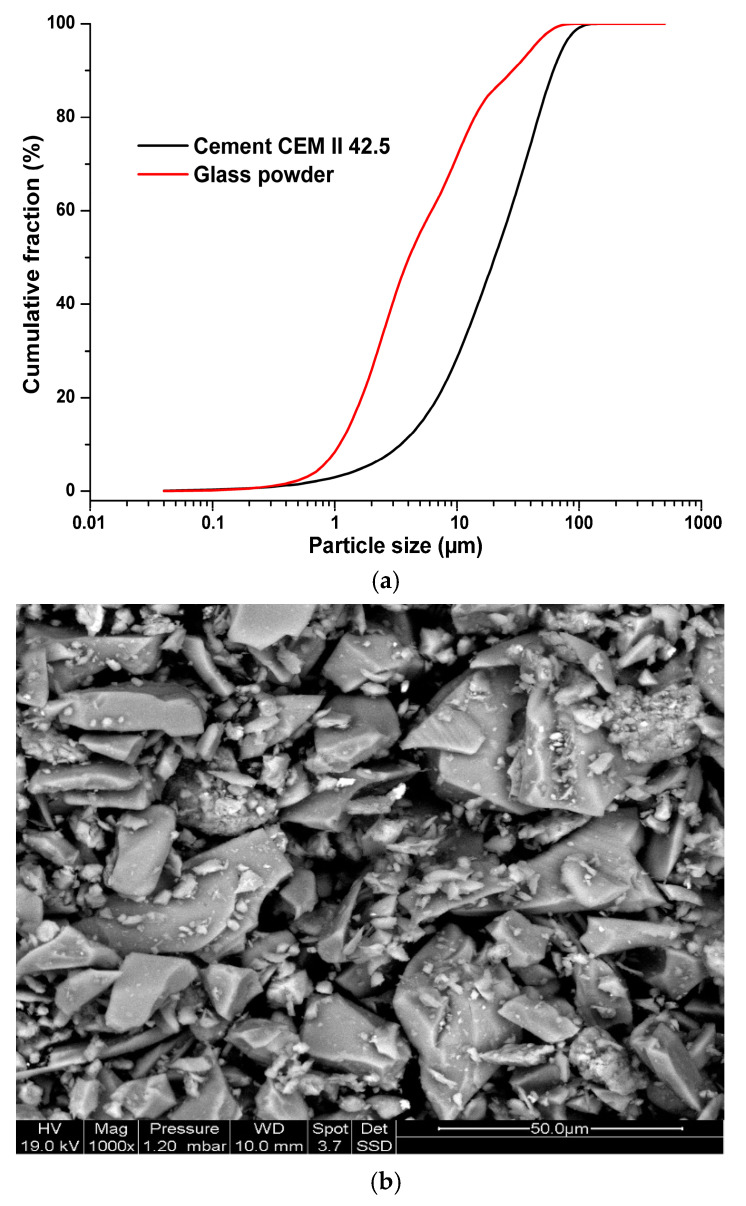
Physico-chemical characteristics of glass powder used: (**a**) grain size distribution, (**b**) particle shape and (**c**) chemical components distribution.

**Figure 2 materials-14-01872-f002:**
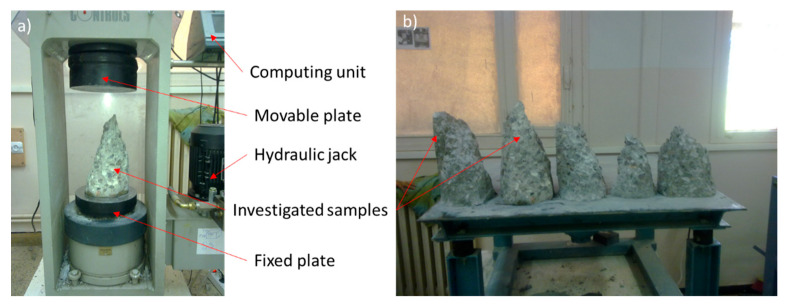
View of the investigated samples: (**a**) in the test stand during the test and (**b**) failure models after the tests.

**Figure 3 materials-14-01872-f003:**
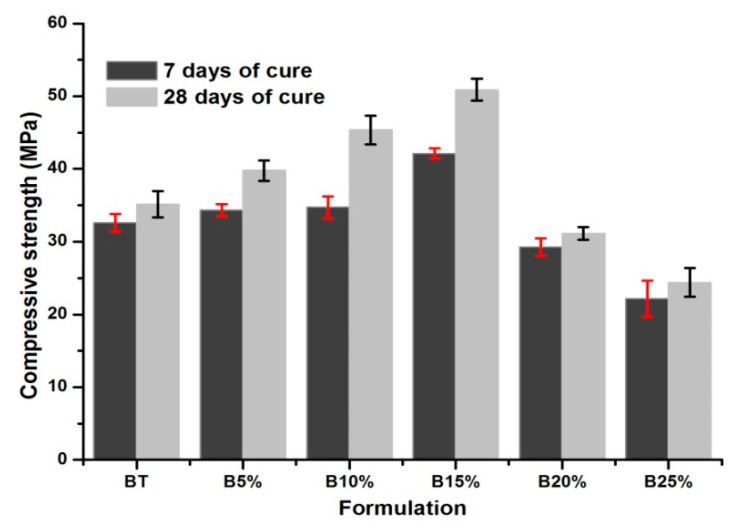
Compressive strength of mixtures at 7 and 28 days.

**Figure 4 materials-14-01872-f004:**
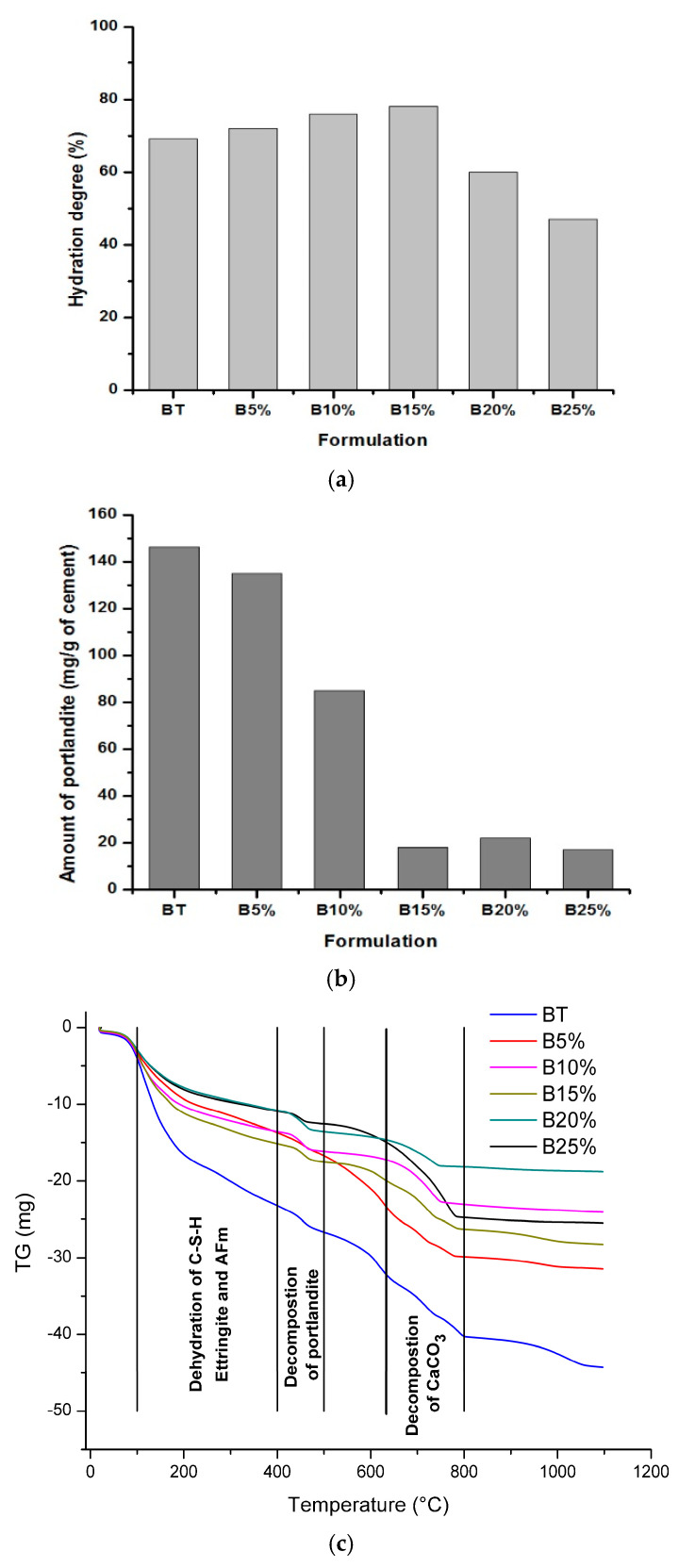
(**a**) Hydration degree, (**b**) amount of portlandite at 28 days and (**c**) thermogravimetric (TG) curves of studied concretes.

**Figure 5 materials-14-01872-f005:**
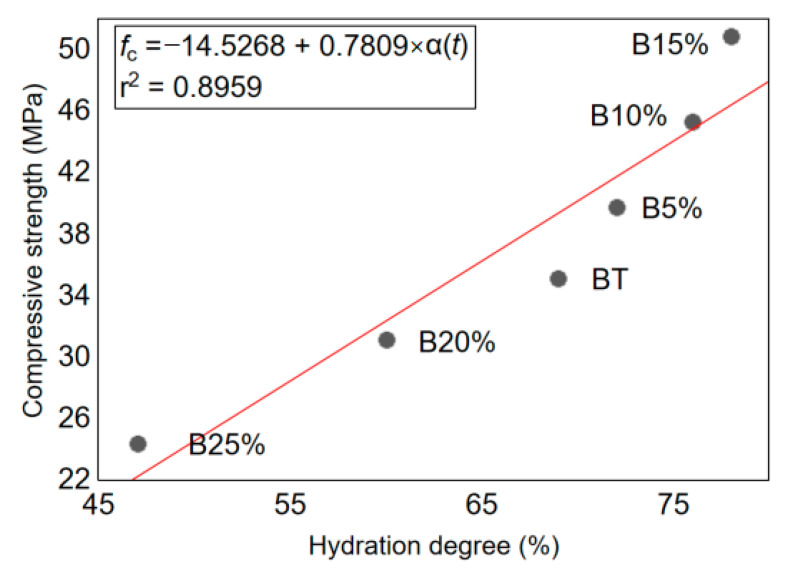
Correlation between the hydration degree and the compressive strength of the concrete with waste glass powder admixture.

**Figure 6 materials-14-01872-f006:**
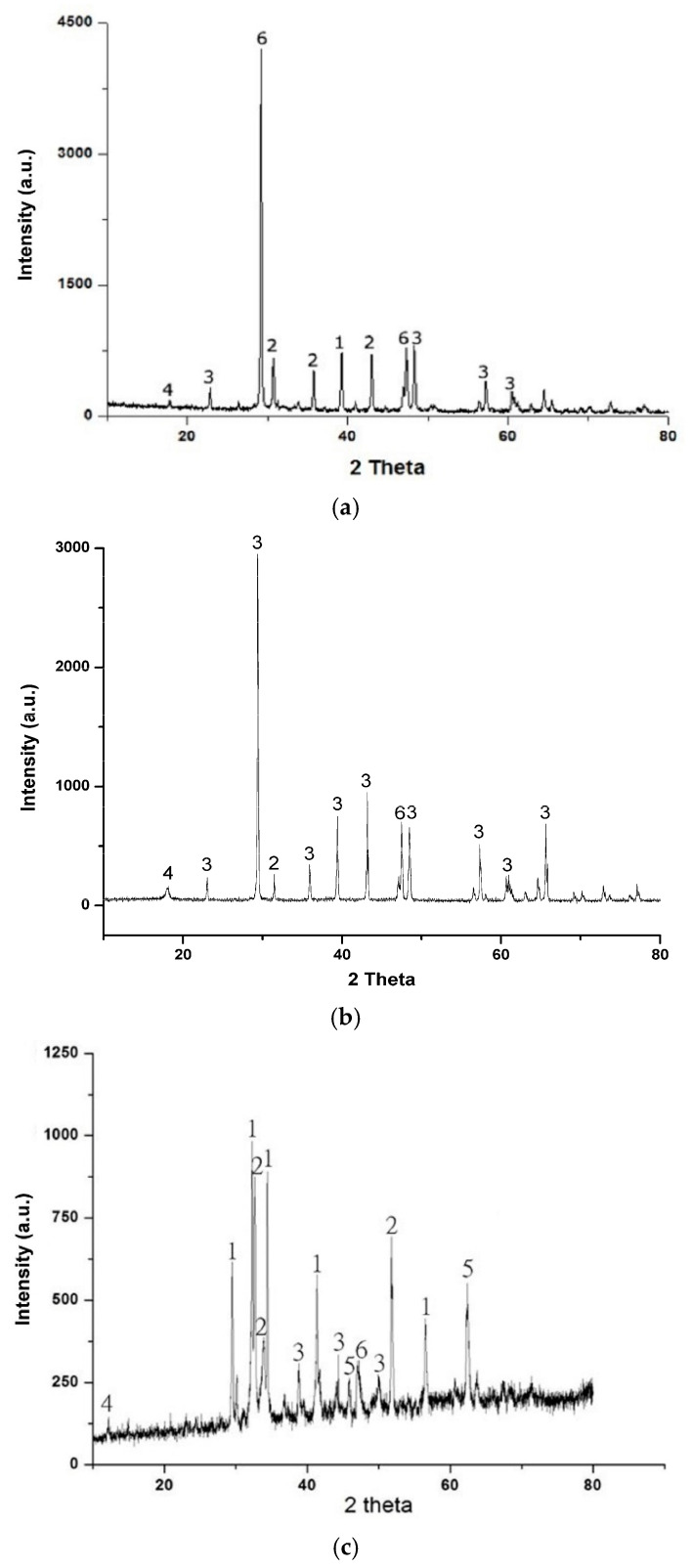
X-ray diffraction (XRD) spectrogram of concrete: (**a**) BT, (**b**) B15% and (**c**) B25%.

**Figure 7 materials-14-01872-f007:**
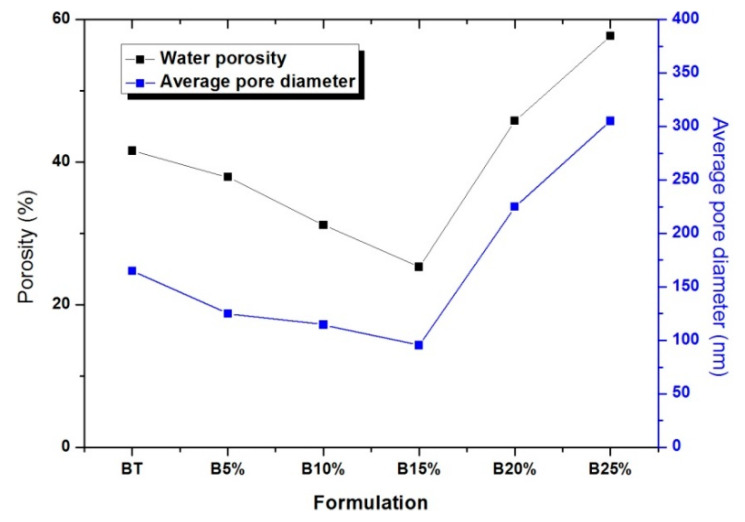
Evolution of water porosity and average pore diameter (obtained by BET technique) of concrete mix design studied.

**Figure 8 materials-14-01872-f008:**
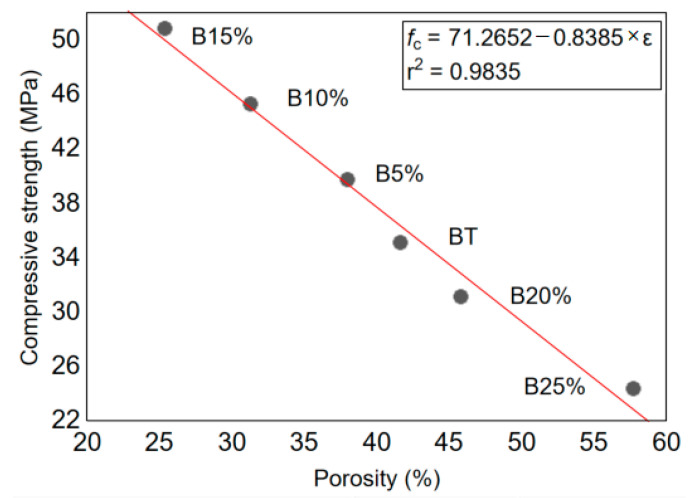
Correlation between the porosity and the compressive strength of the concrete with waste glass powder admixture.

**Table 1 materials-14-01872-t001:** Exemplary studies covering the topic of usage waste glass aggregate as an eco-friendly admixture.

Authors	Mixture Content	Analysed Properties	Main Findings
Nassar et al. [6]	Milled waste glass as a 20% by weight replacement of cement, and 50% and 100% replacement of aggregate. Mixture type: *Traditional concrete*	Slump test, density, strength (compressive and flexural), sorption,	Higher water absorption, improvement in pore system characteristics, enhance durability such as sorption, chloride permeability, and freeze–thaw resistance
Islam et al. [7]	Glass powder as a 10–25% of cement replacement. Mixture type: *Mortar*	Flow test, compressive strength, cost analysis	Increase of strength, reducing costs.
Schwarz et al. [17]	Glass powder as 5%, 10% and 20% cement replacement by mass in mortar. Mixture type: *Mortar*	Strength activity index, effective conductivity, degree of hydration,	Increase of strength activity index, decrease of effective conductivity, decrease of hydration degree
Soliman et al. [18]	Glass powder as a replacement of cement in UHPC (from 0 to 50% of replacement by mass). Mixture type: *Ultra High Performance Concrete*	Compressive strength, heat flow, workability, hydration process	Increase of compressive strength, heat flow reduction, greater workability, slower hydration process
Mirzahosseini et al. [19]	Clear glass and green glass as a 25% replacement of cement Mixture type: *Cement Paste*	Chemical shrinkage, heat of hydration, absorption, compressive strength	Increase of chemical shrinkage, increase of heat of hydration, increase of compressive strength,
Małek et al. [20]	Glass cullet as a replacement of granite aggregate up to 20%. Cement content was not reduced. Mixture type: *Lightweight concrete*	Slump cone, porosity, pH values, bulk density, strength (compressive, flexural, split, tensile), elasticity of modulus, Poisson coefficient	Slump cone reduction, slight reduction of the density, increase of strength (compressive, flexural, split tensile strength)
Chung S-Y et al. [22]	Crushed glass or expanded glass as a full replacement of natural sand (0–4 mm). Cement content was not reduced Mixture type: *Traditional concrete*	Porosity, thermal conductivity, density, strength (compressive and flexural)	Thermal conductivity reduction, slight reduction of porosity, increase of strength (compressive and flexural)
Aliabdo et al. [23]	Glass powder as 5–25% cement replacement Mixture type: *Traditional concrete*	Thermo-gravimetric analysis, strength (compressive and tensile), slump test, density, sorption	Increase of compressive and tensile strength, decrease of absorption, Increase of density,
Kim et al. [24]	Glass powder as a 10% and 20% cement replacement Mixture type: *Traditional concrete*	Slump test, strength (compressive and flexural), porosity	Increase of compressive strength, porosity reduction,
Yousefi et al. [25]	Expanded glass as a 50% and 100% replacement of natural aggregate. Cement content was not reduced. Mixture type: *Mortar*	Flow test, density, water absorption, thermal insulation	Density reduction, increase of water absorption, compressive strength reduction, heat transferring rate reduction

**Table 2 materials-14-01872-t002:** Physico-chemical characteristics of cement and glass powder.

**Cement CEM II 42.5**							
Compounds	SiO_2_	Al_2_O_3_	Fe_2_O_3_	CaO	MgO	SO_3_	K_2_O
(%)	19.83 ± 0.87	6.21 ± 0.24	3.12 ± 0.08	60.52 ± 1.31	0.94 ± 0.09	1.02 ± 0.06	0.01 ± 0.00
Compounds	Cr_2_O_3_	Na_2_O	Ignition loss = 2.41 according to NF EN 196-2				
(%)	-	0.05 ± 0.01	SSA = 3310 cm^2^/g Density = 3.10				
C3S = 49.41%		C2S = 19.85%		C3A = 11.18%		C4AF = 9.48%	
**Glass Powder**							
Compounds	SiO_2_	Al_2_O_3_	Fe_2_O_3_	CaO	MgO	SO_3_	K_2_O
(%)	74.15 ± 0.93	2.12 ± 0.06	0.75 ± 0.09	5.63 ± 0.03	1.45 ± 0.14	-	0.15 ± 0.02
Compounds	Cr_2_O_3_	Na_2_O	Ignition loss = 0.85				
(%)	0.19 ± 0.03	8.44 ± 0.21	SSA = 4530 cm^2^/g Density = 3.53				

**Table 3 materials-14-01872-t003:** Energy-dispersive X-ray spectroscopy (EDS) quantification of glass powder.

Element	O	Na	Mg	Al	Si	Cr	K	Ca	Fe	S
(%)	45.45 ± 0.39	8.3 ± 0.2	2.04 ± 0.01	2.54 ± 0.09	31.85 ± 0.36	0.03 ± 0.02	0.65 ± 0.06	7.22 ± 0.36	1.87 ± 0.12	0.02 ± 0.01

**Table 4 materials-14-01872-t004:** Mix designs of one cubic meter of concrete in kg/m^3^.

Designation	Cement	Glass Powder	Water	Sand 0/3	Gravel 3/8	Gravel 8/16
BT	350	0	175	713	314	628
B5%	332.5	17.5	175	714	314	629
B10%	315	35	175	715	315	629
B15%	297.5	52.5	175	716	315	630
B20%	280	70	175	716	315	630
B25%	262.5	87.5	175	717	316	631

**Table 5 materials-14-01872-t005:** Slump of fresh concretes.

Designation	BT	B5%	B10%	B15%	B20%	B25%
Slump (mm)	115	105	90	85	70	65

**Table 6 materials-14-01872-t006:** Mineralogical compounds detected by using X’Pert High Score software.

Legend	Compound Name	Reference Pattern
1	Calcium silicate	00-042-0551
2	Larnite	00-033-0302
3	Calcite	00-005-0586
4	Ettringite	00-041-1451
5	Tricalcium aluminate	00-038-1429
6	Portlandite	00-004-0733

**Table 7 materials-14-01872-t007:** Cost and decision ranks of the cementitious composites with waste glass powder.

Mixture	Cement	Glass Powder	Cost	Cost Rank	Compressive Strength	Compressive Strength Rank	Total Rank
[-]	[kg]	[kg]	[Euro/m^3^]	[-]	[MPa]	[-]	[-]
BT	350	0	100	6	35.16	4	10
B5%	332.5	17.5	98.8	5	39.80	3	8
B10%	315	35	97.6	4	45.35	2	6
B15%	297.5	52.5	96.4	3	50.89	1	4
B20%	280	70	95.2	2	31.15	5	7
B25%	262.5	87.5	94	1	24.41	6	7

## Data Availability

Data sharing not applicable.

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
