# Peer review of "Mechanical and Microstructural Properties of Ordinary Concrete with High Additions of Crushed Glass"

_materials, 2021, doi:10.3390/ma14081872_

Round 1

Reviewer 1 Report

The manuscript investigates the effect of partial replacement of cement with crushed glass on the mechanical properties and the microstructure of ordinary concrete. The topic is relevant to sustainable development and circular economy implementation. However, the subject of the paper, namely the used of glass as a partial surrogate of cement, the test methods and the approach adopted for the investigation, have already been quite used by the scientific community.

Thus, for the sake of scientific accuracy as well as to constitute an actual contribution for science advance, the paper’s introduction needs to be better contextualized and supported with literature references as well as the methods and results' presentation and discussion need to be improved. In addition to this generic comment that should be considered, the suggestions for authors are presented below:

Ln 24/184: The alkaline compound does not catalyse the pozzolanic reaction. In fact, the portlandite (CH) is a reactant.

Ln 25: Typically, the pozzolans do not accelerate the cement hydration reaction, and this conclusion is not clearly shown in the paper, and it is even written the contrary somewhere. Improve the presentation of the results so that it is unequivocal.

Ln 27: Improve the porosity results presentation so that the conclusion drawn in this sentence becomes clear.

Ln 50-57: This presented aim of the paper is not innovative. It is also not innovative the mix design method nor the techniques used to evaluate the properties nor the microstructure. Thus, better contextualize the subject and describe the intended contribution of the paper for the science advance.

Ln 73: EBSD refers to the sensor used for the images’ visualization. To perform local chemical analysis, another technique must be involved. Please specify. Is it Energy Dispersive Spectroscopy (EDS)?

Ln 77: How many local chemical analyses have been made so that you can assume that they are representative of the material (even though glass is mainly silicious for sure)? This comment is also related to the results presented in Table 2.

Table 1: Clarify the test method used to determine the Ignition Loss.

Ln 111- 115: The treatment of the TGA results need to be better explained so that the interpretation of the results can also be understandable. For example: How was determined the bound water content, the anhydrous cement mass? and water mass related to the cement? Which is the range of temperature considered for portlandite (CH) loss? Which method was used for CH quantification (stepwise or tangential)?

Section 2.2.4-2.2.5: Better explain the methods used to determine the total porosity (%) and the average pore diameter.

Ln 148-149: Specify the method used to determine the SSAs

Ln 161-163: Explain why the contribution of the pozzolanic products for the compressive strength has not been considered?

Ln 166: Since the explanation is supported on the 'decreasing rate' of the reactions, it seems that after a given curing time, the same strength is reached. Thus, why was the curing time not increase? On the other hand, in the Abstract is written that the glass accelerates the hydration...

ln 173-177:  The understanding of Figure 5 requires the details referred for lines 111-115 (above); Explain the details of the determination of ratio CH/cement; Present the TGA and/or DTA curves.

Figures 6-8: In all the legends of figures, adopt the same number for the same compound. Confirm if in figures 6 and 7 the 1st peak of compound number 3 is the CH or CaCO3; Add to the manuscript a table with the reference of ICDD cards of the compounds identified as well as with the main peaks.

Ln 210-212/224: Clarify the method used to perform XRD quantitative analysis.

Ln 215: Fig 5b) reveals the presence of CH in B15%, at 28 days. Correlate this difference among the results.

Ln 220-221: Identify the uncrystallized zones through the angle. Better explain the sentence.

Reviewer 2 Report

The article sounds more like a report than scientific article. In this form, it is not written well enough for publication in such a respectable journal.

In introduction, which has to be rewritten, it is necessary to present an overview ordinary concretes with high additions of crushed glass in concrete and mortar and briefly present the results of interest. At the end of the Introduction or as a separate chapter, it is necessary to explain what is missing in those researches, which will explain the novelty and scientific contribution of this research.

References:

“Gao, T., Shen, L., Shen, M., Chen, F., Liu, L., Gao, L. Analysis on difference of carbon dioxide emission from cement production 296 and their major determinants. Journal of Cleaner Production 2015, 103, 160-170. https://doi.org/10.1016/j.jclepro.2014.11.026 297

Turcry, P., Oksri-Nelfia, L., Younsi, A., Aït-Mokhtar, A. Analysis of an accelerated carbonation test with severe preconditioning. 298 Cement and Concrete Research 2014, 57, 70-78. https://doi.org/10.1016/j.cemconres.2014.01.003 “

should be replace with the newer one, such as:

„Sheheryar, M.; Rehan, R.; Nehdi, M.L. Estimating CO2 Emission Savings from Ultrahigh Performance Concrete: A System Dynamics Approach. Materials 2021, 14, 995. https://doi.org/10.3390/ma14040995“

The reviewer is convinced that there are much more references in different journals, not only in Construction and Building Materials (references [17-23]:

“Soliman, N.A., Tagnit-Hamou, A. Development of ultra-high-performance concrete using glass powder - Towards ecofriendly concrete. Construction and Building Materials 2016, 125, 600-612. https://doi.org/10.1016/j.conbuildmat.2016.08.073

Yu, X., Tao, Z., Song, T.Y., Pan, Z. Performance of concrete made with steel slag and waste glass. Construction and Building Materials 2016, 114, pp. 737-746. https://doi.org/10.1016/j.conbuildmat.2016.03.217

Kim, I.S., Choi, S.Y., Yang, E.I. Evaluation of durability of concrete substituted heavyweight waste glass as fine aggregate. Construction and Building Materials 2018, 184, 269-277.https://doi.org/10.1016/j.conbuildmat.2018.06.221

Cassar, J., Camilleri, J. Utilisation of imploded glass in structural concrete. Construction and Building Materials 2012, 29, 299-307. https://doi.org/10.1016/j.conbuildmat.2011.10.005

Aliabdo, A.A., Abd-Elmoaty A.E.M., Aboshama A.Y. Utilization of waste glass powder in the production of cement and concrete. Construction and Building Materials 2016, 124, 866-877. https://doi.org/10.1016/j.conbuildmat.2016.08.016

Elaqra, H.A., Abou-Haloub, M.A., Rustom, R.N. Effect of new mixing method of glass powder as cement replacement on me-345 chanical behavior of concrete. Construction and Building Materials 2019, 203, 75-82. https://doi.org/10.1016/j.conbuildmat.2019.01.077

Mohammadinia, A., Wong, Y.C., Arulrajah, A., Horpibulsuk, S. Strength evaluation of utilizing recycled plastic waste and recycled crushed glass in concrete footpaths. Construction and Building Materials2019, 197, 489-496. https://doi.org/10.1016/j.conbuildmat.2018.11.192 “

After just a quick search on the web, a lot of articles regarding similar research topics can be found.  For example:

Małek, M.; Łasica, W.; Jackowski, M.; Kadela, M. Effect of Waste Glass Addition as a Replacement for Fine Aggregate on Properties of Mortar. Materials 2020, 13, 3189. https://doi.org/10.3390/ma13143189”

From the sentences:

„Research goal presented in this work is to design a sustainable ordinary concrete containing a waste glass powder (GP) as partial cement replacement with dosage optimization, based on the performed laboratory tests. On the other hand, the applicational goal is to design a product made of cement and waste glass powder which is attractive from 58 economical and sustainable point of view, and can be used in cementitious composites“

it is not acceptable description of the novelty and scientific contribution of the article since there are a lot of studies with waste glass powder.

Authors should choose references that will indicate the problem they want to solve. After such a presentation, they should clearly emphasize the contribution of the article.

It should be deleted some of the self-citations:

“Czarnecki S., Shariq M., Nikoo M., Sadowski L., An intelligent model for prediction of the compressive strength of cementitious 323 composites with ground granulated blast furnace slag based on ulstrasonic pulsce velocity measurements, Measurement 2021, 324 172, 108951. https://doi.org/10.1016/j.measurement.2020.108951

Toubal Seghir, N.; Benaimeche, O.; Krzywiński, K.; Sadowski, Ł. Ultrasonic Evaluation of Cement-Based Building Materials Modified Using Marble Powder Sourced from Industrial Wastes. Buildings 2020, 10, 38. https://doi.org/10.3390/build-327 ings10030038

Chowaniec, A.; Sadowski, L.; Zak, A. The chemical and microstructural analysis of the adhesive properties of epoxy resin coatings modified using waste glass powder. Applied Surface Science 2020, 144373. https://doi.org/10.1016/j.apsusc.2019.144373“

since there are lot of other articles with cement replacement with wastes.

SEM analyses are required because the microstructure is extremely important since larger percentages of cement are replaced.

The article must include full stress-strain curves during compression and load-deflection curves during bending.

The authors must describe in detail with sketches the failure modes, i.e. crack patterns of concrete specimens and their evolution.

More detailed conclusion regarding obtained result should be provided.

Reviewer 3 Report

The current study investigates addition of crushed glass collected from waste as a partial replacement for cement that is commonly used in concretes. The aim here is to optimise the mixing of concrete contents and for that the authors test six different mixtures by changing the percentage of crushed glass in the concrete mix from 5-25%. The authors then carry out several mechanical tests to understand the effect of crushed glass content on the mechanical and thermal performance of the concrete. The authors report that addition of 15% crushed glass resulted in improve concrete strength due to higher degree of hydration.

Line 16 “Optimisation of mix design” What does the authors mean here? Please combine with the following sentence in the abstract or remove it.

The author mention the word optimisation in the abstract but it is not clear what technique was used for the optimisation study?

The introduction needs more work, the authors must summarise the past work carried out on this topic of using waste materials such as crushed glass in concrete applications, discuss what was done and what were the main findings from those studies, then explain how this work brings new knowledge and significance to the field.

The authors are encouraged to prepare a table as shown below and discuss it in detail in the introduction (6-10 studies if you can find that many):

Concrete name

Mixture content

Analysed parameters

Main findings

Ref

A

Crushed glass type 1

Shear

Tensile

Bending..

B

Crushed glass type 2

….

C

Data in Table 1 if not measured by the authors then it must be referenced

How important it is that the crushed glass particle size is similar to that of the cement? Does it improve mixing for example?

Combine figures 1-3 in one larger figure

Why did the authors choose these mixing percentages, are they based on previous literature, or recommendation from industry or just systematic analysis by increasing the crushed glass content in cement up to certain level? Please explain, for example why not going up to 35% what is the recommended maximum limit for such applications.

The authors are encouraged to add some figures showing their test setup and the machines and equipment used in the study

So it is clear from figure 4 that addition of 15% crushed glass gives highest compressive strength but the authors did not explain why this is the case?  

“This improvement of  strength can be attributed to the significant specific surface of glass powder, which con-tributes to the filling effect” can you support this with references and report on past studies which agree or disagree with your findings. I am not convinced that this is the only reason for improved compressive strength?

Again same for section 3.3 explain why best results are found when using 15%. Please explain and support your justifications by citing past studies or literature and explain any possible mechanisms or phenomena that might influence the concrete performance based on the crushed glass content.

From line 207-226 combine in one paragraph rather than bullet points.

Round 2

Reviewer 1 Report

The authors made a major revision to the paper and in some aspects, they improved it remarkably.

However, not all changes improved the paper. In fact, the authors recognized the relevance of some comments but they did not find always a good way to overcome the questions. For instance, the authors recognized that the paper needed to be more contextualize regarding the previous studies published as well as that its innovation should be clarified. However, the authors were not properly neither systematized nor comprehensive in performing this task but it was presented as if the literature survey was completed and able to support the contribution of the paper to science. This need to be more accurate.

Moreover, some of the comments were not properly approached. For instance, the last three.

In addition, some of the 'new' results included are not accurately analysed/discussed.
Consider the comments in the file attached.

Not all the references are correctly referred to in the paper.

The English language and the style are needing revision.

Reviewer 2 Report

The authors significantly improve the manuscript and accepted all suggestions.

Author Response

Authors woul like to thank the Reviewer for the revision.

Reviewer 3 Report

All questions were answered

Author Response

(The authors gave the same response as above.)

Round 3

Reviewer 1 Report

The paper again improved significantly. Even though, it would have been recommended less assertively in some of the claims for which the scientific support is not available or mentioned.

The authors did not properly approach the following comments (already included in the previous review report) that are important to consider for the sake of scientific accuracy:

>>Table 2 and 3: In the previous review report, it was mentioned that the chemical composition of glass powder included in Table 2 and Table 3 are not coherent. The authors recognized in the cover letter that the sample is not homogenous nor the analytical techniques are the same. However, the authors did not include this justification in the manuscript. In addition, besides although the techniques are not the same in order to be adequate they have to provide similar results. This might require making a significant local chemical analysis by EDS.  The two tables do not make sense as presented without the difference in the results explained, as the authors did. The pragmatic recommendation now is not to consider table 3.

>> Lines 24, 275-277, 284-288, 313-315, 329-330, 436-439: Chromium is either being mentioned either to explain the improvement in the microstructure and the properties as well as to the reverse. As referred to in the previous review report, indicate the chromium threshold value from which the chromium slows down the hydration reaction and specify the samples under study for which it applies.

>> Lines 348-366, 440-443 - see comment in the attached files
